# Ultrahigh Performance Liquid Chromatography–Electrospray Ionization Tandem Mass Spectrometry Method for Qualitative and Quantitative Analyses of Constituents of *Corydalis bungeana* Turcz Extract

**DOI:** 10.3390/molecules24193463

**Published:** 2019-09-24

**Authors:** Miao Tian, Chunjuan Yang, Jing Yang, Hongrui Dong, Lu Liu, Yixuan Ren, Zhibin Wang

**Affiliations:** 1Key Laboratory of Chinese Materia Medical (Ministry of Education), Heilongjiang University of Chinese Medicine, Harbin 150040, China; CreatorA@163.com; 2Department of Pharmaceutical Analysis and Analytical Chemistry, College of Pharmacy, Harbin Medical University, No. 157 Baojian Road, Nangang District, Harbin 150081, China; chunjuanyang@126.com (C.Y.); donghongrui422@163.com (H.D.); a1534064875@163.com (L.L.); renyixuan1218@163.com (Y.R.); 3Analytical Department, Johnson & Johnson, 199 Grandview Road, Skillman, NJ 08558, USA; jjsprite09@gmail.com

**Keywords:** alkaloids, *Corydalis bungeana* Turcz, qualitative and quantitative, tandem mass spectrometry, ultrahigh-performance liquid chromatography

## Abstract

In this study, the constituents of a *Corydalis bungeana* Turcz extract were qualitatively analyzed using gradient elution with a mobile phase of 0.2% acetic acid and acetonitrile. We obtained comprehensive insight into the constituents of *C. bungeana* Turcz extracts through the quantitative analysis of 14 compounds by comparison with authentic reference standards, and tentatively identified an additional 44 compounds through electrospray ionization mass spectrometry (ESI–MS) and tandem MS detection. The separation was successfully achieved using an Agilent SB-C_18_ column (1.8 µm, 150 × 2.1 mm; Agilent, Santa, CA, USA). A tandem quadrupole spectrometer was operated in either full-scan mode or multiple reaction monitoring (MRM) for the qualitative and quantitative analyses of the constituents, respectively. Validation data (inter-day and intra-day combined) for accuracy and precision ranged from −4.80% to 4.73%, and 0.30% to 4.97%, respectively. An ultrahigh performance liquid chromatographic–ESI–tandem MS (UHPLC–ESI–MS/MS) method for qualitative of *C. bungeana* Turcz (*C. bungeana*) extract and the quantification of 14 alkaloids, namely, A–N, was developed and validated. Quantitative analysis involved gradient elution with a mobile phase of 0.1% acetic acid and methanol for 45 min. The separation was successfully achieved using a Waters SB-C_18_ column (1.8 µm, 100 mm × 2.1 mm, Waters, Milford, Massachusetts, USA). The repeatability and stability of the method also met USFDA criteria with CV values lower than 5%. The limit of detection of the 14 alkaloids ranged from 9.74 to 13.00 ng/mL, whereas the linearities of the standard curves were between 0.9991 and 0.9995. In total, 15 commercial samples from 11 different sources were analyzed.

## 1. Introduction

The global market share of traditional Chinese medicine (TCM) has been growing, given their relatively few side effects in contrast to modern therapeutics. *Corydalis bungeana* Turcz, belonging to the Papaveraceae family, is a perennial herb grown in parts of China [1]. The dried whole plant, also considered *C. bungeana* in TCM, is included in the Chinese Pharmacopeia (2015 Edition) [2]. *C. bungeana* has long been used in China [3] for the treatment of different ailments such as hordeolum [4], icteric hepatitis [5], and hemorrhoids [6]. *C. bungeana* has many constituents including alkaloids, steroids, and flavonoid glycosides [7]. Among these components, the alkaloids are the most abundant active ingredients. Modern pharmacological studies have shown that isoquinoline alkaloids of *Corydalis incisa* have anti-inflammatory [4], antinociceptive [8], antibacterial [9,10], and other related activities. All 14 alkaloids mentioned above are the major constituents of *C. bungeana* that are responsible for its various activities. 

Although the acceptance of herbal remedies is increasing worldwide, the problem of quality control remains. Various analytical methods, including colorimetry [11], high-performance liquid chromatography (HPLC) with UV detection [12,13,14], and thin-layer chromatography (TLC), have been applied to the qualitative or quantitative analysis of alkaloids in vitro. However, the lack of standard compounds and information are two key problems frequently encountered in identification work. Therefore, ultrahigh-performance liquid chromatography mass spectrometry (UHPLC–MS) is the best choice for qualitative analysis of the chemical components in plants. Especially given the absence of standards, UHPLC–MS has been commonly applied in the identification of minor components in TCMs. UHPLC–tandem MS (UHPLC–MS/MS) methods are fast and more selective and sensitive, and have become the primary choice for both qualitative and quantitative analyses of complex samples, such as TCMs. For example, Hossain developed a rapid UHPLC–MS/MS method to identify and quantify steroidal alkaloids of potato, and the method can be used in other potato varieties [15]. In our previous research, we isolated several alkaloids from *C. bungeana* extracts. In that work, we achieved full characterization of the chemical constituents of *C. bungeana*. Overall, the complexity of TCMs has been determined via the development of analysis techniques, leading to the separation and detection of more previously unidentified components. Owing to the high sensitivity of the multiple reaction monitoring (MRM) technique, liquid chromatography coupled with the triple quadrupole MS method has been extensively applied in quantitative analysis. To the best of our knowledge, no qualitative and quantitative methods for examining *C. bungeana* using UHPLC–ESI–MS/MS have been published. In the current research, we developed an UHPLC–ESI–MS/MS method for inferring the structure of the main constituents of *C. bungeana* and the validated UHPLC–ESI–MS/MS method with a positive MRM mode was used for the determination of 14 central alkaloids.

## 2. Results

### 2.1. Method Optimization 

The results showed that the alkaloid content in the 70% ethanol extract was higher than in other experimental solvents or extraction methods, as determined using a method combining quantitative analysis and qualitative evaluation. The mobile phase was adjusted and optimized to achieve optimal separation and ionization efficiency. Acetic acid (0.2% for qualitative; 0.1% for quantitative) was used to adjust the pH value of the mobile phase. To achieve appropriate chromatographic separation, an Agilent SB-C_18_ column (1.8 µm, 150 mm × 2.1 mm; Agilent, Santa, CA, USA) was used to identify constituents of *C. bungeana*, and a Waters SB-C_18_ column (1.8 µm, 100 mm × 2.1 mm, Waters, Milford, MA, USA) was used for the quantitative analysis. The gradient ratio and column temperature were also optimized, and the gradient elution programs are listed in Appendix A. The column temperature of both methods was then set to 35 °C.

### 2.2. Qualitative Analysis

Through UHPLC–ESI–MS/MS analysis, we found that the positive ion mode is more suitable for alkaloid detection, and fragment ions of sufficient abundance can be stably detected. The aim of optimizing collision energy is to obtain suitable ion fragments using MS/MS. Over 67 peaks were detected within 50 min in the MS chromatogram of the 70% EtOH extract in both negative and positive ion modes (Figure 1). The compounds were tentative identified on the basis of UHPLC retention time as listed in Table 1 and Table 2. The possible chemical composition of the 44 compounds were inferred by means of rigorous study of their MS and MS/MS spectra, and we compared literature data and information from Agilent PCDL manager (Agilent, Santa, CA, USA). The deduced chemical structures of 32 constituents in positive ion mode are displayed in Figure 2. The quantification of alkaloids A–N was also confirmed by spiking samples with reference compounds that were available in our laboratory [16].

#### Fragmentation Regularity

Alkaloids are the main active compounds in *C. bungeana*. Approximately 42 different alkaloid compounds derived from *C. bungeana* have been reported so far, with ~105 alkaloid compounds reported from the species of the genus *Corydalis* [17]. The alkaloids are mainly isoquinoline alkaloids. Based on previous research, the isoquinoline alkaloids gained from this herb can be divided into the following types; protopine (A), proberberine (B), aporphine (C), benzophenanthridine (D), benzylisoquinoline (E), and others [16,18]. According to this classification, compounds 3, 12, and 23 are type A; compounds 4, 7, 9, 10, 11, 18, 20, 24, 26, 37, and 38 are type B; compounds 5 and 30 are type C; compounds 13, 14, 22, 25, 27, 33, and 39 are type D; and compound 8 was the only type E isoquinoline alkaloid isolated in this study. The characteristics of the pattern of fragmentation for the different types of isoquinoline alkaloids vary regarding the pattern that is followed. Accordingly, the chemical structure of some alkaloid compounds can be deduced on the basis of their characteristic fragments.

To study the fragmentation patterns of these constituents and obtain more complete information about the product ions, the collision energy (CE) was examined at various values (20, 30, and 40 eV). Taking the MS/MS data of components 12, 4, 5, 6, 27, 22, and 8 (the representative compounds of different types of alkaloids) as an example, the structures were determined as follows.

The [M + 1]^+^ ion of compound 12 at *m/z* 354 corresponds to its molecular weight. When analyzing the fragments of this compound, we found that the fragmentation pattern of this alkaloid corresponds to a retro-Diels–Alder (RDA) mode of fragmentation (*m/z* 206 and 165) [19]. The loss of fragments corresponding to peaks at *m/z* 308 and 320 is consistent with a loss of H_2_O or methoxyl. We also found that additional peaks at *m/z* 188, 194, and 91 from ESI are characteristic of protopine. This demonstrated that compound 12 is protopine, which was also confirmed by spiking with a reference compound. Compounds 3 and 23 had similar fragmentation pathways. 

The [M + H]^+^ ions of compounds 4, 7, 9, 18, 26, and 38 that eluted at 6.6, 83.1, 10.9, 19.4, and 44.7 min, respectively, were observed to undergo the RDA fragmentation reaction. The C ring opened due to the presence of a double bond in the C ring opposite to the B ring. Two fragment ions were obtained from the precursor. Taking compound 4 as an example, two fragment ions (*m/z* 178 and 151) were obtained, and the RDA fragmentation reaction always occurred as type B dissociation [20,21].

For aporphine-type alkaloids with a secondary amine ring, the initial fragmentation involved a loss of methylamine [M − NCH_3_]^+^ (*m/z* 296) to form a three-membered cyclic ring. The unique losses of 31 and 30 Da, due to the loss of CH_3_OH (*m/z* 265) with the subsequent loss of OCH_3_ (*m/z* 235), led to the formation of a stable three-membered ring structure. The [M + 1]^+^ ion at *m/z* 328 of compound 5 was observed to follow a similar fragmentation pathway as described above. Hence, compound 5 was deduced to be corytuberine [22]. Compound 30 also had a similar fragmentation pathway.

The MS/MS spectra of compound 8 exhibited a pseudo-molecular ion at *m/z* 300, with ion [M − 30]^+^ (*m/z* 269) or [M − 123]^+^ (*m/z* 177) obtained after the elimination of a nitrogen methyl group and subsequent loss of a benzyl group. We proposed that compound 8 is *N*-methylcoclaurine, which is type E. Further supporting evidence was found by the existence of ions corresponding to *m/z* 191 and 107 in the fragmentation pattern. The C-1 and 1a in *N*-methylcoclaurine are prone to breakage and the resulting formation of fragment ions. The double bonds in the adjacent rings tend to rearrange, and the electrons in the vicinity of the nitrogen atoms are influenced by the electrophilic effect of two double bonds, which leads to the final cleavage [23].

Corynoline and acetylcorynoline are representative type D compounds and the main components of *C. bungeana*. The ESI mass spectra of compound 22 produced a characteristic quasi-molecular ion of 368. At certain collision energy and fragmentor voltage, the electrons around the nitrogen atoms were influenced and fragmented into smaller molecules (*m/z* 177 and 135). The unique losses of 15 and 76 Da occurred due to the loss of CH_3_ (*m/z* 352) with the subsequent loss of 2OCH_2_ (*m/z* 292), which was also confirmed through addition of a reference standard. Compound 22 was found to be corynoline, compound 25 was found to be acetylcorynoline, and compound 39 was found to be 8-oxocorynoline.

In conclusion, a total of 32 alkaloids were identified in positive ion mode. The identification of chromatographic peaks in positive ion mode mainly depended on the literature data and the information from Agilent PCDL manager. In total, we confirmed the identities of 12 phenolic acids, as listed in Table 1 and Table 2. The proposed fragmentation patterns of the characteristic compounds in *C. bungeana* mentioned above are listed in Figure 3. This is the first time corytuberine, jateorhizine, corydaline, and coryptopine have been identified in *C. bungeana*, and the first report of bassianin, bilatriene, berberrubine, and worenine in Papaveraceae.

### 2.3. Quantitative Analysis

#### 2.3.1. Optimization of the Condition

At the beginning of the present study, we separated 14 alkaloids, (Acetylcorynoline(A), 8-oxocorynoline (B), Corynoline (C), Tetrahydropalmatine (D), Protopine (E), Palmatine (F), Columbamine (G), Jateorhizine (H), Berberine (I), Worenine (J), Sanguinarine (K), Berberrubine (L), Coptisine (M), and Z23 (N), from *C. bungeana* using UHPLC–ESI–MS/MS. We selected MRM mode to monitor MS/MS ion transitions avoiding the co-eluting peaks. The qualitative results indicated that these alkaloids are the major constituents in this plant and the optimized parameters are listed in Appendix A. Figure 4 depicts the typical MRM chromatograms of 14 compounds. 

#### 2.3.2. Linearity

Stock solutions, including 14 reference standards (Acetylcorynoline(A), 8-oxocorynoline (B), Corynoline (C), Tetrahydropalmatine (D), Protopine (E), Palmatine (F), Columbamine (G), Jateorhizine (H), Berberine (I), Worenine (J), Sanguinarine (K), Berberrubine (L), Coptisine (M), and Z23 (N), were configured and diluted to seven different concentrations for the preparation of calibration curves. The linear regression equations, correlation coefficients, and ranges of calibration curves for the listed alkaloids are display in Table 3. The calibration curves for actual standards were linear, with correlation coefficients larger than 0.9991. The limits of detection (LODs) and limits of quantification (LOQs) were 3 and 10 times the noise level, respectively. The LODs and LOQs for all standard analytes were in the range of 4.87 to 6.50 and 9.74 to 13.00 ng/mL, respectively, revealing that this method is sensitive for the quantitative analysis of the major components of *C. bungeana*.

#### 2.3.3. Precision and Accuracy

The standard solutions with low, medium, and high concentrations were determined at the optimum conditions, and intra- and inter-day variations were chosen to measure the precision of the method. The intra- and inter-day precisions were within 0.30% and 4.97%, respectively. The accuracies ranged from −4.80% to 4.73%. The intra-and inter-day precisions and accuracies are displayed in Table 4.

#### 2.3.4. Repeatability and Stability

The repeatability and stability of the method were also validated for each analyte. The RSDs were less than 2.88% and 4.83%, respectively. Our experimental method was found to have high repeatability and stability, as shown in Appendix A.

#### 2.3.5. Recovery

As shown in Table 5, the recovery rate of the 14 standards varied from 92.11% to 108.84% (RSD ≤ 4.60%). These results verified the high recovery of this method.

#### 2.3.6. Sample Detection

The proposed method was used to quantify the 14 alkaloids in *C. bungeana* from 15 sources (S1–S15). As shown in Table 6, quantities of A–N in *C. bungeana* varied depending on their origin. For example, the alkaloid content, except for jateorhizine originating from Hebei Baoding and Zhejiang, was significantly higher in Hebei Anguo than in Gansu, Henan, and other places. In addition to Hebei Baoding and Zhejiang, the quantities of total alkaloids in plants from Jilin Changchun were higher than in plants from other provinces. The alkaloid contents among the herb samples varied widely according to differences in growing conditions and postharvest storage, and further research is necessary for complete characterization and understanding of these differences.

According to the quantity profiles of the 14 alkaloids, the analytical results demonstrated that *C. bungeana* from S1 (Appendix A) has the greatest similarity with S9 and S11, in accordance with the actual contents of the alkaloids from the 15 sources. *C. bungeana* from Bozhou in Anhui province was the most optimal material for extraction of S2, S13, and S15 based on their higher abundance. However, S3 and S6 differed in other sources to a lesser extent, which illustrated that *C. bungeana* from Tongjiang in Zhejiang province and Jiangsu were lower quality as some of the alkaloids, such as D, F, H, and K, could not detected in these sources. The cluster analysis results provide a foundation for the selection of raw materials of *C. bungeana* and guarantee its quality for efficient clinical usage.

## 3. Materials and Methods 

### 3.1. Reagents and Materials

*C. bungeana* was collected in different provinces in China from the period of August 2015 to March 2016, and was identified by Professor Zhenyue Wang of Heilongjiang University of Chinese Medicine (Harbin, China). The 15 sources and other information about *C. bungeana* are listed in Appendix A. Acetylcorynoline (A), 8-oxocorynoline (B), corynoline (C), protopine (E), and Z23 (N) with 98% purity, as determined using UV, MS, NMR, and HPLC analysis, were isolated from *C. bungeana* provided by Harbin Medical University (Harbin, China). The alkaloids tetrahydropalmatine (D), palmatine (F), columbamine (G), jateorhizine (H), berberine (I), worenine (J), berberrubine (L), and coptisine (M) were purchased from the Chengdu Pufei De Biotech Co., Ltd. (Chengdu, China). The chemical structures of the 14 alkaloids are listed in Appendix A. HPLC-grade solvents containing methanol, water, acetic acid, 2-propanol, and acetonitrile were purchased from Sigma-Aldrich (Wicklow, Ireland). Ultrapure water was prepared using a MilliQ water purification system (Millipore, Molsheim, France).

### 3.2. Preparation of Standards

The crude extract of *C. bungeana* (10 kg) was powdered, mixed, and extracted with 95% ethanol (200 L, 3 × 1 h) three times, and then filtered. The total extract was evaporated under reduced pressure at 50 °C, and then dissolved in 100 L water, passed through D101 macroporous resin, and eluted with water including 30%, 60%, and 90% (*v/v*) ethanol in succession. The samples were subject to chromatography on silica gel and on Sephadex LH-20. The 60% and 90% ethanol elute yielded protopine, corynoline, Z23, 8-oxocorynoline, and acetylcorynoline. The structures of these alkaloids were confirmed by comparing their MS and NMR (^1^H-NMR and ^13^C-NMR) spectral data with those reported [24]. HPLC–DAD and UHPLC-MS were applied to analyze the percentage of total peak area, which showed that their purities exceeded 98%.

### 3.3. Sample Preparation

Dried powder (1 g) of *C. bungeana* was refluxed with 70% ethanol for 1 h. The extraction was repeated twice, and the extracts were combined and dissolved in 10 mL water. The aqueous solution was then extracted three times with 30 mL CHCl_3_ to remove the lipophilic impurities. The total alkaloids were extracted with CHCl_3_ twice. The CHCl_3_ extracts were combined and evaporated to dryness, and the extracts were combined and dissolved in 25 mL with methanol and stored in a refrigerator. The herbal extract was passed through a syringe filter of 0.22 µm and the filtrate was then injected for UHPLC–ESI–MS/MS analysis.

### 3.4. Instrumentation and Operation Conditions

#### 3.4.1. Optimization of the UHPLC System 

The analysis was performed on a series 1290 UHPLC instrument (Agilent, Santa, CA, USA) coupled with a 6430 QQQ-MS mass spectrometer with an ESI interface consisting of a binary pump G4220A, autosampler G4226A, and TCC G1316C module. Two separation methods were employed in the analysis. (1) Qualitative analysis was performed to analyze the chemical constituents of *C. bungeana*. To retain as many chemical constituents as possible, we adjusted the separation conditions using an Agilent SB-C_18_ column (1.8 µm, 50 mm × 2.1 mm) (Agilent Technologies, Santa Clara, CA, USA) at 35 °C with the flow rate of mobile phase set to 0.35 mL/min. The mobile phase consisted of 0.2% acetic acid in water (A) and acetonitrile (B). A linear gradient elution program was developed as follows; 0–5 min: 30% B; 5–10 min: 30%–31% B; 10–15 min: 31% B; 15–18 min: 31%–33% B; 18–20 min: 33% B; 20–21 min: 33%–34% B; 21–28 min: 34% B; and 28–50 min: 52% B. The sample injection volume was 10 µL. (2) A quantitative assay was simultaneously performed to determine the 14 alkaloids in *C. bungeana*. Chromatographic separation was conducted on a Waters C_18_ column (ACQUITY UPLC® HSS T3, 1.8 µm, 2.1 mm × 100 mm, Waters, Milford, MA, USA) at 35 °C. The mobile phase consisted of 0.1% acetic acid in water (A) and methanol (B). The eluting conditions were optimized as follows; isocratic at 30% B (0–5 min), linear gradient from 30% to 31% B (5–10 min), 31% B (10–15 min), 31% B to 33% B (15–18 min), 33% B (18–20 min), 33% to 34% B (20–21 min), 34% B (21–28 min), 34% to 45% B (28–40 min), and 45% B (40–45 min). The flow rate was 0.35 mL/min, the autosampler was set to 10 °C, and the sample injection volume was 10 µL. The gradient elution programs for the mobile phase for qualitative and quantitative analyses are listed in Appendix A, respectively.

#### 3.4.2. Optimization of MS System

The ionization of the two methods was achieved using an ESI interface. The MS parameters were as follows: capillary voltage of 4.5 kV, dying gas flow of 11 L/min, source temperature of 100 °C, and desolvation temperature of 350 °C. In qualitative and quantitative analyses, the nebulizer gas pressure was 40 and 45 psi, respectively. 

MassHunter workstation software version B.02.00 (Agilent, Santa, CA, USA) was used to control the system, acquire data, and conduct qualitative and quantitative analysis. In the qualitative analysis, the TIC spectra were obtained in positive and negative ion modes from *m/z* 50 to 800. We measured protonated and deprotonated molecules to determine the precise molecular masses of different compounds in positive and negative ion modes, respectively. An Agilent PCDL Manager (Agilent, Santa, CA, USA) equipped with the TCM database was applied for qualitative analysis. We used MRM to quantify different steroidal alkaloids in the quantitative assay. The specific parameters of the 14 alkaloids are provided in Appendix A.

### 3.5. Validation of UHPLC–ESI–MS/MS Quantitation

#### 3.5.1. Preparation of the Standard Solutions

For LC–MS/MS analysis, the 14 reference compounds were accurately weighed, placed separately into 10 mL volumetric flasks, and dissolved in methanol to create reference stock solution. All solutions were stored at 4 °C and brought to room temperature before use. Calibrated reference working solutions were freshly prepared by appropriate dilution of the mixed stock solution. Their respective concentrations are as follows (in μg/mL); A = 6.50, B = 11.60, C = 5.11, D = 5.07, E = 4.87, F = 5.65, G = 5.575, H = 5.00, I = 5.40, J = 5.55, K = 5.00, L = 5.525, M = 5.00, and N = 5.34 L. The stock solution was then further diluted with methanol to obtain reference stock solutions, producing final concentrations of 6.50 to 3250 ng/mL for A, 5.80 to 2900 ng/mL for B, 5.11 to 2555 ng/mL for C, 5.07 to 2535 ng/mL for D, 4.87 to 2435 ng/mL for E, 5.65 to 2825 ng/mL for F, 5.60 to 2788 ng/mL for G, 5.00 to 2500 ng/mL for H, 5.40 to 2700 ng/mL for I, 5.55 to 2775 ng/mL for J, 5.00 to 2500 ng/mL for K, 5.53 to 2763 ng/mL for L, 5.00 to 2500 ng/mL for M, and 5.34 to 2670 ng/mL for N. The QC(Quality control) samples were prepared at three different concentration levels: high QC (1625, 1450, 1278, 1268, 1218, 1413, 1394, 1250, 1350, 1388, 1250, 1381, 1250, and 1335 ng/mL), medium QC (325, 290, 256, 254, 244, 283, 279, 250, 270, 278, 250, 276, 250, and 267 ng/mL), and low QC (13.00, 11.60, 10.22, 10.14, 9.74, 11.30, 11.15, 10.00, 10.80, 11.10, 10.00, 11.05, 10.00, and 10.68 ng/mL for compounds 1–14, respectively.

#### 3.5.2. Calibration Curve and Limit of Detection

The calibration curves were created through plotting the peak area obtained in MRM mode against the concentration of each analyte. The limit of detection (LOD) and limit of quantification (LOQ) were regarded as the analyte concentration with a signal-to-noise ratio (SNR) of approximately 3 and 10, respectively.

#### 3.5.3. Precision and Accuracy

Intra-day and inter-day precision for each analyte at three different QC concentrations were determined using six replicates on the same day (intra-day) and over three consecutive days (inter-day). The accuracy was calculated by the relative error (RE) of the measured mean value deviated from the nominal value. 

#### 3.5.4. Recovery

Recovery was used to further evaluate the accuracy of the method. Known amounts of each standard solution at different concentration levels (high, middle, and low) were added to known amounts of sample (Zhejiang, China). The samples were analyzed as described above to determine the recoveries of the 14 *C. bungeana* samples. Six parallel samples were prepared and the RSD of the recoveries was calculated.

#### 3.5.5. Repeatability and Stability

Five samples from the same batch (Zhejiang) were extracted and determined to measure the repeatability of this method (*n* = 6). The stability of the alkaloids in *C. bungeana* extract was assessed by repeated injections at 0, 1, 2, 4, 8, and 12 h to detect the peak area of these alkaloids. The percentage of RSD over the time points during this period was used to measure the stability of the compounds.

### 3.6. Qualitative Analysis and Quantitative analysis

For exploring this method, the extracted ion mode and product ion mode were used, and an ESI source was operated in positive and negative modes simultaneously during qualitative analysis. The MRM mode was then employed for quantitative analysis. Finally, 15 batches of *C. bungeana* from 11 different cities in China were compared using this new method.

## 4. Conclusions

In this research, qualitative and quantitative methods were combined to evaluate the quality of the chemical constituents of *C. bungeana*. An UHPLC–ESI–MS/MS method was developed for qualitative and quantitative analyses of the *C. bungeana* extract. More than 67 peaks were detected in the 70% EtOH extract, and the chemical structures of 44 constituents were deduced according to their MS/MS spectra, fragmentation patterns, and other information. We also employed this method to determine the identities of 14 alkaloids in this plant. This is the first time that 14 alkaloids were simultaneously analyzed. The new method was found to have suitable precision and accuracy, and acceptable repeatability and stability. The results indicate that the components of TCMs can be quantified clearly and that the quality control can be improved using updated analysis methods based on modern technology.

## Figures and Tables

**Figure 1 molecules-24-03463-f001:**
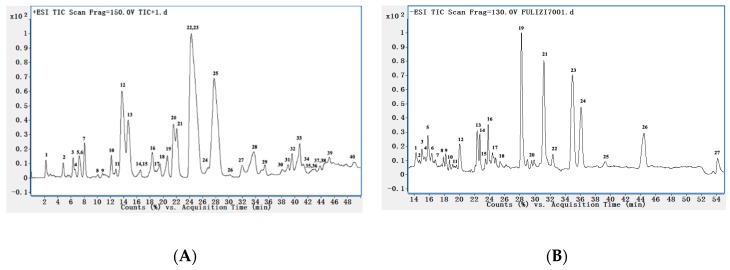
The total ion chromatograms (TIC) of *C. bungeana* extract in positive (**A**) and negative (**B**) ionization mode.

**Figure 2 molecules-24-03463-f002:**
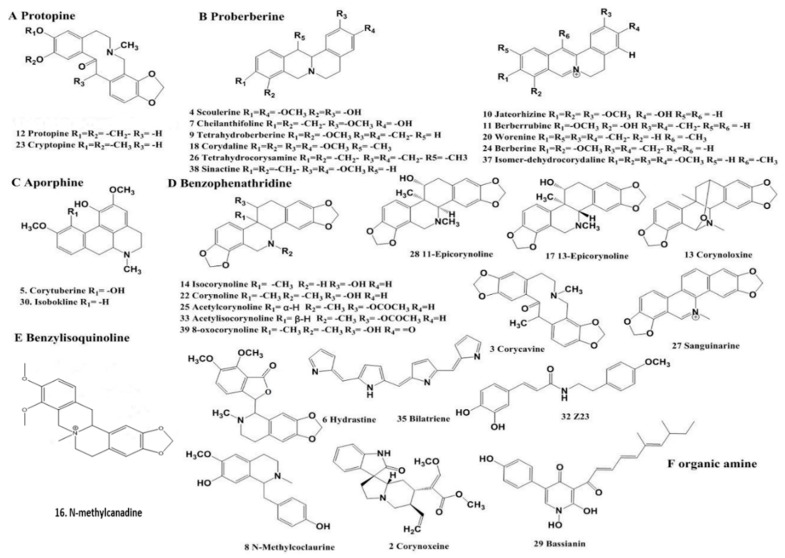
The chemical structures of constituents identified in extract of *C. bungeana*.

**Figure 3 molecules-24-03463-f003:**
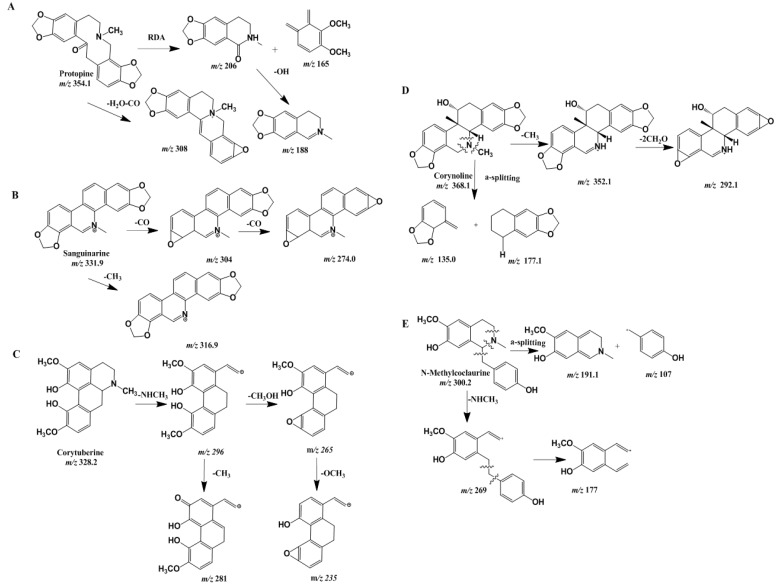
The proposed fragmentation pattern of representative alkaloids in *C. bungeana*.

**Figure 4 molecules-24-03463-f004:**
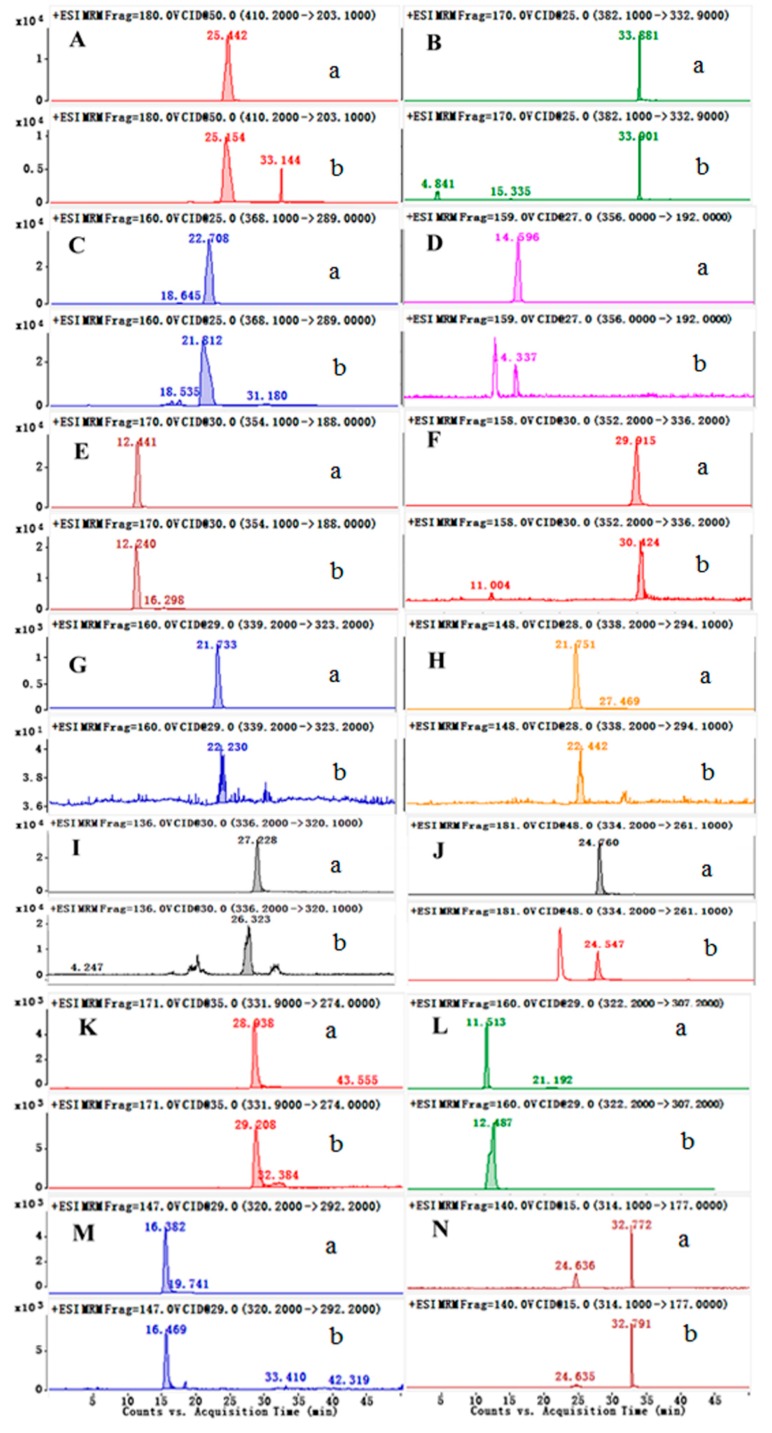
*C. bungeana* UHPLC-MRM chromatograms. (**a**) Fourteen alkaloids (Acetylcorynoline (**A**), 8-oxocorynoline (**B**), Corynoline (**C**), Tetrahydropalmatine (**D**), Protopine (**E**), Palmatine (**F**), Columbamine (**G**), Jateorhizine (**H**), Berberine (**I**), Worenine (**J**), Sanguinarine (**K**), Berberrubine (**L**), Coptisine (**M**), and Z23 (**N**) reference solutions. (**b**) Test solution of *C. bungeana*.

**Table 1 molecules-24-03463-t001:** Precursor and product ions of the constituents in the 70% EtOH extract of *C. bungeana* in ultrahigh performance liquid chromatography–electrospray ionization mass spectrometry–tandem mass spectrometry (UHPLC–ESI^+^–MS/MS) experiments.

No.	T_r_ (min)	Calcd Mass [M *−* H]^+^	MS/MS Fragments *m*/*z*	Tentative Identification
1	2.1	301 [M + X]^+^	Unknown	
2	4.9	382 [M]^+^	178; 265; 222; 162; 207	Corynoxeine
3	6.4	368	277; 190; 90; 44; 307;	Corycavine
4	6.6	328	178; 150; 264	Scoulerine
5	7.4	328	296; 235; 281	Corytuberine
6	7.5	384	364; 303; 58	Hydrastine
7	8.1	326	178; 151; 91; 311; 283	Cheilanthifoline
8	10.1	300	191; 177; 107	N-Methylcoclaurine
9	10.9	340	208;114;	Tetrahydroberberine
10	12.2	338	322; 294	Jateorhizine
11	12.7	323	307; 279; 250; 197	Berberrubine
12	13.9	354	188; 149; 247; 320; 91	Protopine
13	14.8	366	317; 206; 146	Corynoloxine
14	16.6	354	188; 149; 247; 249	Isocorynoline
15	16.5	426	Unknown	Lythranidine
16	18.4	354 [M]^+^	190; 247;1 88; 174; 166	N-methylcanadine
17	18.8	368	176; 289; 319; 174	13-epicorynoline
18	19.4	369 [M]^+^	174; 274; 246; 259	Corydaline
19	20.3	368 [M + X]^+^	Unknown	
20	21.6	334 [M]^+^	261; 291; 2333; 332; 147	Worenine
21	22.0	408	Unknown	Erysothiopine
22	24.3	368	289; 261; 177; 319; 231; 135	Corynoline
23	25.1	369 [M]^+^	336; 190; 188; 158; 181	Coryptopine
24	26.8	336 [M]^+^	320; 292	Berberine
25	27.9	410	204; 246; 162; 332	Acetylcorynoline
26	30.1	338	277; 163	Tetrahydrocorysamine
27	32.0	331 [M]^+^	317; 274; 246; 218; 189	Sanguinarine
28	33.5	368	289; 261; 177	11-Epicorynoline
29	34.8	396	114; 209; 114	Bassianin
30	37.9	312	119	Corytuberine
31	39.1	394	Unknown	Coryincine
32	39.6	314	289; 177; 261; 231; 135; 79	Z23
33	40.7	410	394; 349; 334; 321; 190; 176	Acetylisocorynoline
34	41.2	344[M + X]^+^	Unknown	
35	42.6	299	295; 99	Bilatriene
36	43.1	395[M + X]^+^	Unknown	
37	43.8	366[M]^+^	350; 334; 308; 292	Isomerdehydrocorydaline
38	44.8	340	113	Sinactine
39	45.3	382	333;275;247;189;135	8-oxocorynoline
40	49.2	181[M + X]^+^	Unknown	

**Table 2 molecules-24-03463-t002:** Precursor and product ions of the constituents in the 70% EtOH extract of *C. bungeana* in UHPLC-ESI^−^-MS/MS experiments.

No	T_R_ (min)	Calcd mass [M − H]^+^	MS/MS Fragments *m*/*z*	Tentative Identification
2	14.8	329	279; 223; 212; 194	Tianshic acid
5	15.9	942	923; 733	Soysaponin 1
6	16.5	795 [M+HCOO^−^]^−^	615; 113; 119	Dipsacussaponin L
7	17.3	311	223; 57	Caftaric acid
8	18.0	577	299	Acacia-7-o-β-d-apiose-(1→2)-β-d-glucose
16	23.9	295	277; 195; 171; 183	Tanshinone II A
19	28.2	277	127;59	Linolenic acid
20	29.6	339	177; 133; 105	Aesculin
21	31.3	279		Isolinolic acid
22	32.4	253	223; 195	Daidzein
23	34.9	255		Bupleurynol
24	35.9	281	236; 203; 174; 150	Gloeosteretriol

**Table 3 molecules-24-03463-t003:** Linearity equations, correlation coefficients, linear ranges, and limits of quantitation for the fourteen alkaloids determined.

Compound	Linear Range (ng/mL)	Linearity Equation	R^2^	LOQ (ng/mL)
Acetylcorynoline	13.00–3250	Y = 18.47X + 829.27	0.9992	13.00
8-oxocorynoline	11.60–2900	Y = 1.66X + 57.44	0.9994	11.60
Corynoline	10.22–2555	Y = 60.43X + 1730.90	0.9994	10.22
Tetrahydropalmatine	10.14–2535	Y = 54.83X + 2275.20	0.9991	10.14
Protopine	9.74–2435	Y = 33.34X + 1766.50	0.9991	9.74
Palmatine	11.30–2825	Y = 58.96X + 1811.50	0.9992	11.30
Columbamine	11.15–2788	Y = 15.74X + 943.69	0.9991	11.15
Jateorhizine	10.00–2500	Y = 45.28X + 1907.30	0.9992	10.00
Berberine	10.80–2700	Y = 13.67X + 478.40	0.9995	10.80
Worenine	11.10–2775	Y = 5.64X + 230.93	0.9993	11.10
Sanguinarine	10.00–2500	Y = 14.54X + 500.13	0.9992	10.00
Berberrubine	11.05–2763	Y = 86.92X + 4687.20	0.9991	11.05
Coptisine	10.00–2500	Y = 16.42X + 712.74	0.9994	10.00
Z23	10.68–2670	Y = 1.67X + 44.10	0.9996	10.68

**Table 4 molecules-24-03463-t004:** Intra- and inter-assay accuracy and precision values of the LC–MS/MS method for the measurement of fourteen alkaloids in *C. bungeana* (*n* = 6).

Compounds	Nominal Mass Concentration (ng/mL)	Observed Mass Concentration (ng/mL)	Accuracy (bias %)	Intra-Day Precision (RSD %)	Inter-Day Precision (RSD %)
Acetylcorynoline	13.00	13.00 ± 0.80	0.11	2.42	1.41
	325	312.5 ± 3.90	−3.83	4.21	1.83
	1625	1495.3 ± 62.40	−3.90	1.24	3.82
8-oxocorynoline	11.60	10.60 ± 0.60	−4.80	2.70	1.90
	290	261.8 ± 6.90	−0.60	1.20	1.20
	1450	1462.7 ± 40.00	0.92	2.73	1.32
Corynoline	10.22	10.60 ± 1.00	4.01	4.43	3.74
	255.5	261.8 ± 6.90	2.54	2.62	2.62
	1277.5	1231.3 ± 77.20	−3.63	1.33	1.71
Tetrahydropalmatine	10.14	11.25 ± 0.30	4.73	2.82	3.70
	253.5	275.4 ± 6.50	3.60	2.43	2.62
	1267.5	1235.4 ± 3.10	−2.54	0.30	1.72
Protopine	9.74	10.80 ± 1.60	4.71	4.80	2.41
	243.5	250.8 ± 2.40	3.03	1.02	1.23
	1217.5	1320.5 ± 26.80	3.51	2.03	4.34
Palmatine	11.30	11.20 ± 1.40	−1.02	4.60	3.63
	282.5	285.7 ± 9.70	1.12	3.44	1.82
	1412.5	1331.6 ± 5.40	−1.72	0.40	3.13
Columbamine	11.15	10.80 ± 0.50	−2.82	1.13	3.63
	278.8	273.6 ± 5.50	−1.85	2.00	0.76
	1393.8	1355.7 ± 30.60	−2.70	2.28	3.10
Jateorhizine	10.00	11.60 ± 0.20	4.68	1.60	4.97
	250	244.6 ± 3.40	−2.22	1.37	1.12
	1250	1282.2 ± 13.80	2.55	1.10	4.78
Berberine	10.80	10.14 ± 0.50	−1.13	1.22	3.57
	270	268.9 ± 9.40	−0.43	3.48	0.79
	1350	1342.5 ± 27.20	−0.57	2.03	3.12
Worenine	11.10	10.70 ± 0.90	−4.02	3.78	3.56
	277.5	273.9 ± 12.00	−1.34	4.40	0.78
	1387.5	1364.6 ± 13.00	−1.60	1.04	3.13
Sanguinarine	10.00	11.60 ± 2.30	4.32	4.27	4.89
	250	252.1 ± 11.30	0.99	4.55	4.90
	1250	1226.8 ± 15.50	−1.89	1.30	3.45
Berberrubine	11.05	11.70 ± 0.80	2.30	3.12	3.56
	276.25	276.1 ± 6.60	−0.01	2.42	0.78
	1381.2	1398.3 ± 6.30	1.23	0.45	3.13
Coptisine	10.00	13.20 ± 2.50	3.23	3.70	4.83
	250	266.3 ± 9.10	1.45	3.43	0.78
	1250	1222.9 ± 24.20	−2.20	2.00	2.13
Z23	10.68	11.90 ± 2.20	1.10	1.78	3.56
	267	256.8 ± 6.80	−3.78	2.60	0.78
	1335	1336.5 ± 55.50	1.23	4.22	3.10

Observed mass concentrations are expressed as mean ± SD.

**Table 5 molecules-24-03463-t005:** Recoveries (%) of fourteen reference compounds following spiking the *C. bungeana* (*n* = 3).

Compounds	Original (ng)	Addition (ng)	Detection (ng)	Recovery (%)	RSD (%)
Acetylcorynoline	163	130.0	293.65	100.50	2.43
		162.5	320.89	97.16	3.25
		195.0	359.02	100.52	3.73
8-oxocorynoline	231.62	184.0	421.30	103.08	4.35
		230.0	466.21	101.99	2.20
		276.0	513.62	102.17	3.24
Corynoline	2021.03	1616.7	3643.62	100.36	1.77
		2020.9	4045.78	100.19	1.06
		2425.0	4432.96	99.46	2.88
Tetrahydropalmatine	1034.79	828.3	1850.37	98.46	3.40
		1035.4	2069.05	99.89	2.20
		1242.0	2283.14	100.51	1.41
Protopine	3580.28	2864.0	6439.87	99.84	2.30
		3580.0	7123.05	98.96	4.33
		4296.0	7879.34	100.07	1.77
Palmatine	2103.25	1682.7	3780.40	99.67	2.98
		2103.3	4215.15	100.40	3.21
		2524.0	4662.71	101.40	3.82
Columbamine	15933.82	12746.9	28513.12	98.68	4.30
		15933.7	31850.01	99.89	2.05
		19120.5	35262.08	101.08	1.11
Jateorhizine	1296.32	1036.8	2326.84	99.39	2.88
		1296.0	2607.07	101.38	3.27
		1555.2	2893.62	102.70	4.21
Berberine	3243.26	2594.9	5656.75	93.00	2.51
		3243.6	6374.04	96.52	3.47
		3892.3	7223.13	102.25	3.16
Worenine	728.54	583.2	1326.20	102.48	2.54
		729.1	1445.03	98.27	2.29
		874.9	1615.78	101.41	4.06
Sanguinarine	8653.42	6923.4	15564.26	99.82	1.78
		8654.3	17458.99	101.75	2.26
		10385.2	19321.52	102.72	1.15
Berberrubine	3721.64	2977.6	6962.49	108.84	2.24
		3722.0	7150.28	92.11	4.60
		4466.4	8482.74	106.90	3.28
Coptisine	1698.23	1358.4	3033.27	98.28	3.16
		1698.0	3379.35	99.01	2.32
		2037.6	3742.09	100.31	1.94
Z23	793.85	635.2	1443.83	102.32	2.86
		794.1	1529.76	92.67	4.55
		952.9	1772.56	102.70	1.50

Found = observed value − basal value.

**Table 6 molecules-24-03463-t006:** Mass fractions of the fourteen alkaloids (%) in the *C. bungeana* from 15 different sources (*n* = 3).

Sources	Compounds (%)														
	A	B	C	D	E	F	G	H	I	J	K	L	M	N	Total
1	0.174	0.117	0.334	0.002	0.106	0.002	0.004	0.002	0.022	0.037	0.045	0.010	0.022	0.693	1.570
2	0.175	0.095	0.288	0.001	0.108	0.001	ND	0.001	0.002	0.001	0.002	0.001	0.002	0.213	0.883
3	0.122	0.054	0.181	ND	0.056	0.003	0.003	ND	0.004	0.004	ND	0.004	0.005	0.006	0.440
4	0.001	0.036	0.502	0.001	0.002	ND	0.001	ND	0.001	ND	ND	0.001	0.001	0.004	0.547
5	0.001	0.036	0.600	0.001	0.004	0.001	0.001	0.002	0.001	0.002	0.001	0.001	0.001	0.011	0.654
6	0.006	0.011	0.092	ND	0.024	ND	ND	ND	0.001	0.005	ND	0.001	ND	0.055	0.197
7	ND	0.068	0.710	0.001	0.002	0.001	0.001	0.002	0.001	0.001	0.002	0.002	0.001	0.044	0.826
8	0.014	0.018	0.551	0.001	0.004	ND	ND	0.001	0.001	0.001	0.001	0.001	ND	0.002	0.592
9	0.205	0.132	0.733	0.001	0.099	0.003	0.018	0.001	0.008	0.038	0.067	0.019	0.010	0.431	1.763
10	0.007	0.009	0.430	ND	0.012	ND	ND	0.001	0.001	0.001	0.021	0.001	ND	0.042	0.524
11	0.258	0.193	0.792	0.001	0.121	0.003	0.005	0.002	0.081	0.041	0.054	0.048	0.028	0.328	1.954
12	0.003	0.001	0.649	0.001	0.008	0.001	0.001	0.002	0.001	0.002	0.019	0.003	0.001	0.180	0.866
13	0.029	0.004	0.402	0.001	0.044	0.001	0.001	0.002	0.002	0.021	0.026	0.006	0.001	0.190	0.724
14	0.002	0.016	0.580	ND	0.006	ND	0.001	0.001	0.003	0.001	0.025	0.002	0.001	0.079	0.712
15	0.041	0.009	0.404	0.001	0.049	0.001	0.001	0.001	0.006	ND	0.031	0.016	0.001	0.174	0.730

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
