# Peer review of "Ultrahigh Performance Liquid Chromatography–Electrospray Ionization Tandem Mass Spectrometry Method for Qualitative and Quantitative Analyses of Constituents of Corydalis bungeana Turcz Extract"

_molecules, 2019, doi:10.3390/molecules24193463_

Round 1
Reviewer 1 Report
The manuscript reports qualitative analysis (of what?) was using gradient elution with mobile phase of 0.2% acetic acid and acetonitrile. This is a common HPLC method. "Comprehensive insight into the constituents of Corydalis bungeana Turcz extracts was completed with the identification of 14 compounds by comparing with authentic reference standards." How are the minute interferences co-eluting with other compounds? Although the results are informative, it did not identify the active compounds in the extracts, which are important in drug development. What is the point of tentative identification of 44 additional compounds through electrospray ionization mass spectrometry (ESI-MS). What are the significances of the present findings? More updated references on Corydalis bungeana Turcz should be cited.
Reviewer 2 Report
The paper from Tian et al. reports the application of a UHPLC-ESI-MS-MS method for the qualitative and quantitative analysis of the constituents of the Corydalis bungeana Turcz extract.
The paper is poorly written. Many grammar errors, wrong verbs, unusual sentence construction, incomplete sentences and lapses of style make the text hardly understandable. The presentation is also not careful (several typos are widespread all over the text). Even the abstract deserves a complete revision. As an example, the starting sentence is “Qualitative analysis was using gradient elution with mobile phase of 0.2% acetic acid and 17 acetonitrile” a sentence which is quite unusual as a starting point!
In addition, authors should avoid the use of colloquial sentences such as:
-line 197: Recovery is another vital factor for method validation
-line 211: Cluster analysis is a kind of fuzzy mathematical method.
The instrument here employed is a low-resolution/low accuracy mass spectrometer. For this reason, I suggest removing the numbers after the decimal point in m/z value of all the fragments. In addition, the column “Formula” in Table 1-1 and 1-2 should be removed because elemental composition determination can only be achieved form m/z values obtained with high resolution/accurate mass instrument. Lines 92-94 should be removed accordingly.
While the UPLC analysis shows a good resolution, except for a few compounds for which the authors confirmed the identity of the compounds using standards, the identification of almost all peaks is based exclusively on MS data and is consequently highly tentative and incomplete unless supported with other identification techniques (i.e NMR). Therefore authors should first report as “tentative” all the obtained identification. In addition, they should differentiate between the “unknown” compounds, molecules which weren’t identified because their mass is not reported elsewhere (i.e. peaks 1, 19,34, 36, 40) and molecules which are tentatively identified only according their molecular masses (i.e. 15,21,31) because none MSMS fragment was generated.
Some further comments:
Lines 129-130: rewrite as 6.6, 8.1, 10.9 etc.
Delete the line 140: MRM technology has the advantage of enhanced peak resolution, hydrogen consumption and solvent saving
Line 149: Corynoline and acetylcorynoline are the representative compounds among type D, not E
Table 2: please refer the letters A-N to the right alkaloid
Round 2
Reviewer 2 Report
The authors carefully revised the manuscript, which is now fluent and easy to read.
However, there are still some minor revisions I would like to suggest:
1-As I wrote in the previous letter,
“In addition, authors should avoid the use of colloquial sentences such as:
-line 197: Recovery is another vital factor for method validation
-line 211: Cluster analysis is a kind of fuzzy mathematical method.”
I suggest deleting these two sentence instead of rewriting them.
2-Table 1.1 and 1.2 as in the previous letter:
“ In addition, the column “Formula” in Table 1-1 and 1-2 should be removed because elemental composition determination can only be achieved form m/z values obtained with high resolution/accurate mass instrument.”
Authors have deleted the wrong column. They should arrange the two tables according to this order:
No---tR(min)--- Calcd mass [M−H]+ ---MS/MS fragments m/z----Tentative identification
3-I suggest assigning letters A-N to the corresponding alkaloid. Authors may clarify this correspondence in the caption of Figure 4 and elsewhere in the text in the quantitative analysis paragraph.
Author Response
Dear Editor, I’d like to submit the revised manuscript entitled “Ultra-High Performance Liquid Chromatography–Electrospray Ionization Tandem Mass Spectrometry Method for Qualitative and Quantitative Analyses of Constituents of Corydalis bungeana Turcz Extract” for publication in Molecules.We thank the reviewers for their valuable comments on previous manuscript. We have carefully taken their comments into consideration in preparing our revision. The changes have been clearly highlighted in the revised manuscript. The reply has been described in detail in the responses to the reviewers follow closely of the cover letter. We would like to express our great appreciation to you and reviewers for comments on our paper. Looking forward to hearing from you. Thank you and best regards. Yours sincerely, Zhibin Wang Zhibin Wang, Heilongjiang University of Chinese Medicine, Harbin 150040, Heilongjiang province, China. E-mail: wzbmailbox@hljucm.net Response to reviewers: Thank you very much for your valuable suggestions on our manuscript. According to your opinions, we have revised the manuscript. May I reply to your comments and show you the changes in the revision as follows (all changes have been clearly highlighted in the revised manuscript): Reviewers' comments: Reviewer#2(round 2) Comments and Suggestions for Authors The authors carefully revised the manuscript, which is now fluent and easy to read. However, there are still some minor revisions I would like to suggest: Point 1. As I wrote in the previous letter, “In addition, authors should avoid the use of colloquial sentences such as: -line 197: Recovery is another vital factor for method validation -line 211: Cluster analysis is a kind of fuzzy mathematical method.” I suggest deleting these two sentence instead of rewriting them. Response 1 : Thanks for your comments and we are so sorry about our inappropriate English expression. We have deteled these two sentence. See page 14 line 406 and page 15 line 421 . Point 2. Table 1.1 and 1.2 as in the previous letter: “ In addition, the column “Formula” in Table 1-1 and 1-2 should be removed because elemental composition determination can only be achieved form m/z values obtained with high resolution/accurate mass instrument.” Authors have deleted the wrong column. They should arrange the two tables according to this order: No---tR(min)--- Calcd mass [M−H]+ ---MS/MS fragments m/z----Tentative identification Response 2 : Thanks for your comments. We have arranged the two tables according to this order: No---tR(min)--- Calcd mass [M−H]+ ---MS/MS fragments m/z----Tentative identification. See page 5-7. Table 1-1 and Table 1-2 Point 3. I suggest assigning letters A-N to the corresponding alkaloid. Authors may clarify this correspondence in the caption of Figure 4 and elsewhere in the text in the quantitative analysis paragraph. Response 3 :We are really sorry for the carelessness. According to your suggestion, we have assigned letters A-N to the corresponding alkaloid as following: Acetylcorynoline(A), 8-oxocorynoline(B), Corynoline(C), Tetrahydropalmatine(D), Protopine(E), Palmatine(F), Columbamine(G), Jateorhizine(H), Berberine(I), Worenine(J), Sanguinarine(K), Berberrubine(L), Coptisine(M), Z23(N). And we clarify the caption of Figure 4 (see line 375 page 11)and elsewhere in the text in the quantitative analysis paragraph(see line 367 page 11 and line 380 page 12).
